# Use of Dried Blood Spot Specimens to Monitor Patients with Inherited Metabolic Disorders

**DOI:** 10.3390/ijns6020026

**Published:** 2020-03-26

**Authors:** Stuart J. Moat, Roanna S. George, Rachel S. Carling

**Affiliations:** 1Department of Medical Biochemistry, Immunology & Toxicology, University Hospital of Wales, Cardiff CF14 4XW, UK; 2School of Medicine, Cardiff University, University Hospital Wales, Cardiff CF14 4XW, UK; 3Derriford Combined Laboratory, University Hospitals Plymouth NHS Trust, Plymouth PL6 8DH, UK; roanna.george@nhs.net; 4Biochemical Sciences, Viapath, Guys & St Thomas’ NHSFT, London SE1 7EH, UK; Rachel.Carling@viapath.co.uk; 5GKT School of Medical Education, King’s College, London SE1 1UH, UK

**Keywords:** inherited metabolic disorders, monitoring, treatment ranges, dried blood spots, haematocrit, certified reference material, precision, accuracy, bias

## Abstract

Monitoring of patients with inherited metabolic disorders (IMDs) using dried blood spot (DBS) specimens has been routinely used since the inception of newborn screening (NBS) for phenylketonuria in the 1960s. The introduction of flow injection analysis tandem mass spectrometry (FIA–MS/MS) in the 1990s facilitated the expansion of NBS for IMDs. This has led to increased identification of patients who require biochemical monitoring. Monitoring of IMD patients using DBS specimens is widely favoured due to the convenience of collecting blood from a finger prick onto filter paper devices in the patient’s home, which can then be mailed directly to the laboratory. Ideally, analytical methodologies with a short analysis time and high sample throughput are required to enable results to be communicated to patients in a timely manner, allowing prompt therapy adjustment. The development of ultra-performance liquid chromatography (UPLC–MS/MS), means that metabolic laboratories now have the capability to routinely analyse DBS specimens with superior specificity and sensitivity. This advancement in analytical technology has led to the development of numerous assays to detect analytes at low concentrations (pmol/L) in DBS specimens that can be used to monitor IMD patients. In this review, we discuss the pre-analytical, analytical and post-analytical variables that may affect the final test result obtained using DBS specimens used for monitoring of patients with an IMD.

## 1. Introduction

The use of dried blood spot (DBS) specimens has been used to both screen babies for phenylketonuria (PKU) and to monitor dietary treatment since the 1960s. The introduction of flow injection analysis tandem mass spectrometry (FIA–MS/MS) in the 1990s allowed the accurate detection of numerous biomarkers for inherited metabolic disorders (IMDs) in a single assay, facilitating the expansion of the number of IMDs screened for in the newborn period [1,2]. FIA–MS/MS has also been used to analyse DBS specimens as part of routine monitoring of PKU and other IMDs. Monitoring of patients using DBS specimens is widely favoured due to the convenience of collecting blood from a finger prick onto filter paper in the patient’s home and mailing the sample directly to the laboratory. Furthermore, the sampling procedure is far less invasive than venepuncture and requires a lower volume of blood, which is therefore more suitable and less traumatic for patients that require numerous blood tests. These benefits are particularly advantageous for infants and young children. 

Measurement of analytes in DBS specimens using FIA–MS/MS is rapid (<1.5 min per sample) and results can be communicated to patients in a timely manner allowing prompt therapy adjustment. This is particularly important when monitoring pregnant female PKU patients or when rapid dietary adjustments are required for newly diagnosed patients. However, it should be recognised that FIA–MS/MS methods lack specificity. The absence of chromatographic separation means that specificity is achieved solely by the use of selective reaction monitoring (SRM). Consequently, any isobaric compound with a common daughter ion has the potential to interfere. A further disadvantage is that the analyte(s) of interest is not separated from the sample matrix, which can result in non-specific interferences from phospholipids and salts. In recent years, technology has evolved significantly, and modern mass spectrometers now have the ability to scan faster. This, in conjunction with the introduction of ultra-performance liquid chromatography tandem mass spectrometry (UPLC–MS/MS) allows rapid chromatographic separation. Mass spectrometers now have adequate sensitivity to cope with rapid flow rates, negating the need for sample derivatisation. This means that clinical laboratories now have the capability to routinely analyse biomarkers in DBS specimens with superior specificity and sensitivity using isotope-dilution UPLC–MS/MS. This advancement in analytical technology has led to the development of numerous assays to detect analytes at low concentrations (pmol/L) in DBS specimens that can be used to monitor IMD patients (Table 1). 

One of the main advantages of using DBS specimens is that it allows specimens to be collected in situations where standard blood collection is challenging (problems with sampling, transport and storage). To date, no study has assessed the cost effectiveness of monitoring IMD patients using DBS specimens collected in the home versus conventional blood collection by venepuncture in a clinical setting. However, in a recent cost evaluation study of the use of DBS specimen collection compared to conventional sampling for therapeutic drug monitoring in children, it was shown that switching to DBS sampling was associated with significant cost reductions (societal and health care) [3]. It is important to note that this study did not assess the quality of the analytical results obtained by the two methods of sample collection. 

The perceived benefit of DBS sampling is the assumption that a sub-punch (small cylinder of a fixed diameter) of a defined volume can be obtained from a DBS formed from a non-volumetrically applied blood sample. However, the volume of blood contained in a sub-punch is inconsistent due to numerous factors that have been reported to affect DBS analyte measurements which impact negatively on the quality of the analytical result, leading to imprecision and inaccuracy. These factors do not affect the collection of a fixed volume of liquid whole blood or plasma. It is therefore important to understand and control, where possible, any pre-analytical, analytical and post-analytical variables that may affect the final test result.

## 2. Pre-Analytical Factors

### 2.1. DBS Specimen Collection Devices

Several different bloodspot collection devices are available commercially. These devices can be divided into cellulose cotton-based papers (both untreated and chemically treated) and non-cellulose-based papers. Cellulose-based papers manufactured using cotton linters are most widely used for screening, diagnosis and monitoring. Three cotton filter papers of different qualities are commercially available: PerkinElmer 226 (PerkinElmer, Waltham, MA, USA), Whatman 903, (GE Healthcare, Amersham, UK) and Munktell TFN (Munktell Filter AB, Sweden) and these are widely used for NBS internationally. Filter paper collection devices for capillary blood collection from heel or finger pricks are Class II Medical Devices and should meet international criteria for performance [22]. Filter paper collection devices should be stored as recommended by the manufacturer to ensure that the results of analytical testing are not affected. In addition, these collection devices should not be used after the expiry date printed on the device [22].

The Newborn Screening Quality Assurance Program (NSQAP) at the Centers for Disease Control (CDC) and Prevention (USA) conducts the evaluation of all lots of Food and Drug Administration (FDA) registered collection devices before they are released for NBS and other applications [23,24]. NSQAP annually publishes results of evaluations for each device. Performance characteristics include serum volume (µL) per 3.2 mm disc, absorption time (s), and spot diameter (mm) for a 100 µL volume of haematocrit (Hct)-adjusted whole blood. In general, all filter paper lots comply with the Clinical & Laboratory Standards Institute (CLSI) performance criteria and were found to be homogeneous, where the measured within-spot, within-sheet, and among-sheet variances were within acceptable limits [23,24]. 

The cotton cellulose filter papers, PerkinElmer 226, Whatman 903 and Munktell TFN, have all been pre-qualified by the CDC for NBS. However, only the PerkinElmer 226 and Whatman 903 papers have FDA approval. NSQAP conducted a study of filter paper performance by applying Hct-adjusted (50%), analyte-enriched whole blood (22 NBS analytes added at single levels and in a dose-response series) onto lots of the two FDA-registered collection devices [24]. The matched specimens were tested in house and sent to six reference laboratories for analysis. Results showed overlap in analyte recovery at one standard deviation and varied from 4% to 5%, which was equal to the lot-to-lot variance for each type of filter paper. These results indicated that the performance of the filter papers was equivalent. Continued adherence to the defined criteria of CLSI NBS01 by manufacturers, and voluntary evaluations by NSQAP, produces blood collection devices with consistent performance.

Cellulose cotton-based papers impregnated with various chemicals are also commercially available. However, they are not registered as Class II Medical Devices, and are not CLSI compliant. The Whatman FTA DMPK-A and DMPK-B Cards (GE Healthcare) have been used mainly in the pharmaceutical industry and are treated with chemicals to lyse cells, denature proteins and to facilitate sample stabilization or the recovery of biologically active molecules.

Non-cellulose-based blood collection devices have also been used (Agilent Bond Elut, Dried Matrix Spotting paper (Agilent Technologies, Santa Clara, CA, USA)). It was claimed that the collection device improved the mass spectrometry signal and Hct-independent spot homogeneity. In-house treatment of cards has also been reported for compound specific stability. Other non-cellulose DBS cards comprised of glass microfiber filter papers are also available commercially but are not specifically designed for DBS collection purposes.

CLSI NBS01 details criteria that manufacturers should meet for filter paper used as a whole blood collection device. Defining how the filter paper matrix influences blood collection is important in order to ensure minimal lot-to-lot variability to maintain precision and reproducibility [24]. These international standardisation efforts ensure uniformity of specimen collection, calibrators, quality control (QC) and reference materials for NBS assays. Using DBS specimens for patient monitoring adds additional requirements for the precision and accuracy of analyte recovery. The type of matrix used for calibration and QC materials will influence the analyte recovery. Ideally, methods testing patient DBS should also use DBS calibration and control materials to correct for the filter paper matrix.

### 2.2. DBS Specimen Collection

The process of DBS specimen collection typically involves the application of a non-volumetric amount of blood from a heel or finger prick. The CLSI NBS01-A6 document provides guidance on blood collection onto filter paper for newborn screening programs. However, these procedures can also be applied to the collection of samples from finger pricks used for IMD patient monitoring [22]. DBS collection devices usually have printed broken line circles of a predefined diameter (typically 10–12 mm) for specimen collectors to obtain appropriately sized samples. A single hanging drop of blood of adequate size to fill the circle should be applied to the filter paper as over or under filling the pre-printed circle affects the volume of blood in the sub-punch that is used for analysis [25]. 

When applied to the collection device, the hanging drop of blood disperses by spreading radially across the filter paper, whilst penetrating the porous fibres to fully soak through to the back of the filter paper. Ideally, the distribution of analytes across the filter paper collection device should be constant. However, the plasma component of the blood applied occupies a greater fractional volume of the interior of the filter paper fibres than the erythrocytes and as a result the erythrocytes concentrate at the edge of the bloodspot (which is often visible). Blood should be applied to one side of the card only and the blood should penetrate through to the back of the paper. Multi-layering, multi-spotting and compression of the specimens can adversely affect the concentration of the analytes within the DBS [25]. The CLSI document provides images of good and poor quality specimens. It also discusses the consequences of compressed, poor quality, multi-layered samples and other variables that require control to ensure that good quality specimens are collected for analysis [22]. 

The quality of DBS specimens received into the laboratory for analysis should be assessed subjectively by visual inspection; ensuring that the printed circle is suitably filled with blood; that the blood is spread symmetrically and evenly on both sides of the filter paper. Laboratories should develop standards and guidelines for sample acceptance/rejection criteria. Education and training, which could include the use of online media platforms to reach the target patient population, is of paramount importance to ensure that appropriately sized and good quality specimens are collected [25].

Contamination of the filter papers with disinfectant, wet wipes, infant feeds, faeces and urine should be avoided as this can significantly affect analyte results [26]. Contamination of the filter paper with drinks that contain the artificial sweetener aspartame (methylester of phenylalanine/aspartic acid dipeptide) can lead to artifactually elevated phenylalanine concentrations [27].

### 2.3. Stability of DBS Analytes 

Once the blood has been applied to the filter paper collection device, the DBS specimen should be allowed to completely dry before being transported to the laboratory [22]. The appropriately dried specimen can then be placed in an envelope for transport. It has been demonstrated that drying (N.B—heat sources should not be used) and storage in low humidity conditions improves the stability of DBS samples [22]. The length of time required for air-drying will depend on the local environmental conditions, e.g., room temperature and humidity. DBS drying usually takes from 90 min to 4 h [22].

DBS specimen collection offers great potential for patient monitoring, but the DBS specimens need to be transported to the clinical laboratory for analysis. During transportation, the DBS specimens can be exposed to extreme environmental conditions (e.g., high temperatures, and high humidity). The US FDA found that the temperature in a mail box exposed to the sun can reach 58 °C (136 °F), while the ambient air temperature was 38 °C (101 °F) [28]. These extreme conditions have been shown to directly affect the quantitation of specific amino acids and other metabolites routinely measured for NBS (some of these analytes are also used for patient monitoring). The results indicate that higher sample storage temperatures (25 and 40 °C) and high humidity (75%) significantly influence the short-term stabilities of amino acids (except valine), acylcarnitines, 17-hydroxyprogesterone (17-OHP) and succinylacetone (SUAC) in DBS specimens [29]. 

Most amino acids were shown to be stable in DBS specimens (Whatman 903 and PerkinElmer 226) during a 4 h exposure to sunlight. However, methionine is unstable and showed a relatively high extent of degradation by ~25% [30]. The stability of DBS methionine is particularly poor and at room temperature will degrade by at least 50% in six months. High humidity and ambient temperature will also increase the rate of degradation [30,31]. Results from our own investigations demonstrate that methionine is unstable even at room temperature after 3 days of storage (Figure 1). At 45 °C and 70% humidity, phenylalanine and tyrosine concentrations decreased by 28% and 49% respectively within 24 h [31]. The amino acids tyrosine, leucine, phenylalanine and methionine decrease at a rate of 1.7%, 3.1%, 5.7% and 7.3% per year respectively for the first five years in DBS specimens when stored at room temperature in a dry environment [32]. 

At room temperature, acylcarnitines are slowly hydrolysed to free carnitine (CO), resulting in an increase in CO concentrations [33]. The stability of acylcarnitines is significantly improved if DBS specimens are stored in sealed bags at low temperature. The 2-(2-nitro-4-trifluoromethylbenzoyl)-1, 3-cyclohexanedione (NTBC) is stable in DBS specimens for 45 days at ambient temperature (20–25 °C) [10]. SUAC stored at low humidity lost <5% of initial concentration for up to 30 days. In total, 60% of SUAC is lost during the first 3 days of storage at elevated temperatures and high humidity [29]. Homocysteine is stable up to 30 days at ambient temperature [7]. Methylmalonic acid (MMA) concentrations are stable at room temperature for 7 days, 8 weeks at 4 °C and up to 1 year at −80 °C. However, at higher temperatures, MMA rapidly increases as methylmalonyl-CoA mutase remains active at 5 °C [9]. DBS 17-OHP is reported to be stable at ambient temperature for several years [34,35,36]. No stability data have been published for DBS globotriaosylsphingosine (Lyso-Gb3). 

Evidence from these studies indicate that storing specimens desiccated at −20 or −80 °C can further increase the stability of analytes in DBS samples, often extending the stability from days/months to years. Clinical laboratories offering patient monitoring services using DBS analytes should undertake stability studies under normal and extreme environmental conditions to assess the effect on the measured analyte. The findings of which may help to explain the cause of unexpected/spurious patient results that are occasionally observed when monitoring patients. 

## 3. Analytical Factors

The quantitative analysis of biomarkers in DBS specimens relies on the assumption that the sub-punch used for analysis provides a volumetric measurement that is comparable to a liquid blood sample. However, several factors impact on the quantitative result obtained when analysing a sub-punch taken from a DBS specimen created using a non-volumetrically applied blood sample. The effects of blood volume applied to the collection device, Hct on spot size, analyte distribution across the DBS and extraction recovery have been reported. These factors can all affect the analytical accuracy and precision of the DBS analysis. 

### 3.1. Sample Preparation 

Sample preparation requires a sub-punch to be taken from the DBS or the removal of the entire DBS from the filter paper to enable extraction of the analyte for quantification. Obtaining the sub-punch can be performed manually with simple low-cost handheld punchers, semi-automated instruments which enable bar code reading, sub-punch location and the distribution of samples into a 96 well plate format to enable sample extraction or fully automated robotic punching systems. Various sized puncher heads are commercially available (1.5, 3.2, 4.7, 6 and 8 mm). The 3.2 mm punch is widely used for NBS. A 3.2 mm sub-punch is widely considered to yield on average 3.0 μL of sample, which can pose issues when attempting to achieve pmol/L assay sensitivities, therefore some assays require larger punches or multiple punches for analysis. The limit of detection (LOD) of an assay is determined by using the analyte signal-to-noise ratio (S/N) and is defined as the lowest analyte concentration where its signal can be reliably distinguished from the background instrument noise. Whilst, UPLC–MS/MS instruments are becoming progressively more sensitive, it should be recognised that there is a trade-off between analyte sensitivity and background noise (as the analyte signal increases, so does the background, i.e., the S/N ratio decreases). It is therefore important to optimise assays to increase the S/N ratio and therefore improve analytical sensitivity.

It should also be noted that carry over may occur using the semi-automated/automated systems when punching a specimen that contains the analyte at a high concentration, which has preceded a specimen with a low concentration of the analyte [37]. Carry over may also occur during the analytical process. It is therefore essential that carry-over experiments are performed and where carry over is observed, that blank filter paper sub-punches are analysed between samples.

### 3.2. Sample Volume/Size, Quality and Punch Location

The volume of blood applied to the filter paper influences the size of the DBS formed and also the analytical results. Figure 2 shows a series of DBS formed using varying volumes of whole blood applied to filter paper using a pipette. The typical DBS specimen contains approximately 50 µL of whole blood with an average diameter of 12 mm. The application of a hanging drop of blood to the collection device results in the blood being absorbed into the matrix of the filter paper. The plasma/serum component applied occupies a greater volume of the interior of the paper fibres than that of the erythrocytes (process affected by Hct level) and this leads to the loss of homogeneity across the DBS (chromatographic effect), with the erythrocytes being concentrated at the edge of the DBS. This loss of homogeneity results in increased concentrations of analytes associated with erythrocytes in the peripheral sub-punches relative to central sub-punches taken from the DBS [25,38,39,40].

The size (i.e., volume of blood applied to the filter paper) and quality of the DBS and punch location have a significant impact on the analyte results obtained [25]. Figure 3 demonstrates the effect of sample size (10 µL to 100 µL), punch location (central vs. peripheral) and specimen compression on phenylalanine concentrations in a blood specimen collected from a patient with phenylketonuria as part of routine monitoring. It can be observed that the result obtained for this monitoring sample could range from 931 to 1299 µmol/L, depending on the size of the DBS and punch location. DBS specimens compressed, either incidentally due to being handled/transported when wet or with intent as a means of filling the circle on the blood collection device (see Figure 4A), can lead to falsely low analyte results (Figure 3). 

During method development and validation, the impact of DBS size/volume on analyte concentrations should be assessed by applying increasing volumes of blood onto blood collection devices and measuring both the analyte concentrations and diameters of the DBS. These factors should also be taken into account when interpreting the analytical result. Furthermore, the effect of central and peripheral sub-punches on analyte concentrations should also be assessed as analyte dependent differences in results have been reported [14,25,38,39,40]. The practical application of these results would be to make recommendations on where sub-punches should be taken from the DBS to ensure more reliable monitoring results are obtained, e.g., to take central punches from two separate DBS to obtain the most consistent duplicate results.

It should be highlighted that the diameter of the calibrator, quality control (QC) and external quality assessment (EQA) DBS samples used in different jurisdictions internationally vary significantly. Furthermore, the minimum spot size from patients accepted by testing laboratories may also vary and may be significantly different to the diameter of the calibrator sample, QC sample and EQA sample materials used in the assay.

### 3.3. Effect of Haematocrit

Hct levels can vary significantly between individuals and with disease states. Hct is recognised as a significant factor that affects the characteristics of a DBS specimen (e.g., drying time, homogeneity and extraction of the analyte) as it affects blood viscosity. Specimens with high Hct values result in increased viscosity of the blood, which affects the distribution of erythrocytes and serum across the filter paper collection device. The effect of Hct on DBS size and appearance can be observed visually (Figure 4B)—those DBS with a low Hct are more symmetrical with smooth edges, in contrast to those with higher Hct levels which are smaller, darker and have an uneven edge. Several studies have examined the impact of Hct on the measurement of various analytes and demonstrated an inverse association between DBS size and Hct (DBS size decreases with increasing Hct) [41,42]. This relationship between DBS size and Hct has a significant effect on the quantitation of biomarkers in DBS specimens, when a fixed sized sub-punch is taken. It has been shown that concentrations of amino acids increased with increasing Hct values [38,40]. Leucine, methionine, phenylalanine and tyrosine concentrations were 21%, 20%, 27%, and 8% (respectively) higher in samples with a Hct of 60% vs. 30%) [40]. A similar effect is also seen with C0 (concentrations were 16% lower at a Hct of 30% vs. 60%, when using central sub-punches) [40] and SUAC (−45% bias from added concentration at a Hct of 30% vs. +24% bias at a Hct of 60%) [14]. The effects of Hct and where the sub-punch was taken (i.e., central vs. peripheral) can be additive and even synergistic [40].

At present, there is no assay available to quantify Hct directly from a DBS specimen. The Hct effect can be avoided by analysing the entire DBS. However, for this to be an effective approach, the amount of blood applied must be carefully controlled to enable a defined volume to be applied. New sampling devices are now commercially available that can potentially overcome the Hct and volume effects by analysing the entire DBS instead of taking a sub-punch. These new blood collection devices take up and apply a fixed volume of blood onto the filter paper collection device [43,44,45,46,47,48]. Although these new devices are more expensive than the traditional filter paper blood collection devices, their ability to overcome volume and Hct effects would result in improved analytical performance and should be evaluated for the monitoring of patients with IMDs to ensure treatment decisions are made with increased confidence.

### 3.4. Biomarker Extraction from DBS Specimens and Internal Standards

Various extraction methods for DBS analysis have been described and most analytes measured for IMD patient monitoring are commonly extracted using mixtures of water and organic solvents (mostly methanol or acetonitrile). Optimal extraction requirements will depend on the physical and chemical properties of the analyte and it is essential to understand how the analyte behaves in the DBS specimen following application to the filter paper and during storage. Whilst amino acids and acylcarnitines can be extracted from DBS using 80% methanol, SUAC cannot and requires a different extraction method. One reported method requires the use of hydrochloric acid for elution and ethylacetate for extraction [12]. However, more recently, it has been shown that SUAC, along with amino acids and acylcarnitines, can be extracted using an acetonitrile-water-formic acid mixture containing hydrazine [13]. The concentration of hydrazine has a significant impact on extraction efficiency [49]. However, the extraction and recovery of SUAC is highly variable and method dependent as demonstrated by EQA performance [50]. Some analytes may have a low extraction recovery due to a strong interaction between the analyte and the hydroxyl groups that are present in the cellulose of the filter paper. In addition, interplay of blood volume and Hct may also affect extraction recoveries and mixing or sonication of the DBS may be required.

The quantitation of the analyte is usually achieved with stable isotope dilution and comparison of the analyte signal to the signal of the stable isotope internal standard (IS). The IS should be added in equal concentration to all samples in the analytical batch (including calibrators, QC and patient samples) to compensate for fluctuations in the analyte response during sample preparation and analysis. For DBS assays, it is common practice to add the IS to the extraction solvent. However, this approach cannot compensate for the pre-analytical factors or the sample preparation steps. DBS assay validation should assess the potential differences in the behaviour of the analyte (i.e., extracted from the DBS sample versus the IS added in solution). 

Analyte extraction/recovery is often investigated by comparing the DBS extract result to an aqueous standard solution and the recovery calculated as the ratio of the DBS result to the aqueous standard. The FDA states that “*...recovery of the analyte need not be 100%, but the extent of recovery of the analyte and of the internal standard should be consistent, precise and reproducible. Recovery experiments should be performed by comparing the analytical results for extracted samples at three concentrations (low, medium and high) with un-extracted standards that represent 100% recovery*” [51,52].

Achieving appropriate and reproducible extraction recovery of an analyte(s) from a DBS can be challenging and will vary from one analyte to another. Appropriate validation experiments are required during method development to optimise analyte(s) recovery, e.g., extraction solvents, pH, temperature, requirement of solvation energy (shaking or sonication), and duration of the extraction process. In addition, when undertaking retrospective studies, it is essential to assess the impact of length of storage on analyte extraction to ensure that older samples do not affect the extraction process.

### 3.5. Assay Calibration

A further limitation to the utility of DBS specimens for monitoring IMD patients is the lack of commercially available matrix matched certified reference material (CRM) for the various analytes in DBS specimens on which to standardise laboratory tests. As a result, DBS calibrators tend to be produced in house by collecting blood from a healthy donor or using residual pooled patient samples and adding an aqueous enrichment prior to application onto filter paper. The exact preparation of the DBS calibrator varies between laboratories, e.g., volume of blood added to the filter paper, varying Hct of the specimen, or use of lysed blood specimens. Each factor can affect the measured concentration. More importantly, the analytical method used to assign the DBS calibrator values can also influence the accuracy of the analytical result.

### 3.6. DBS Internal Quality Control

The precision of the DBS assay should be evaluated using appropriate DBS QC samples prepared at a minimum of three different concentration levels (low, intermediate and high). Clinical decision limits should be taken into account when determining the optimum QC concentrations. It is good practice to prepare QC samples and calibrators using freshly collected blood from a healthy volunteer or pooled residual patient samples in order to emulate the clinical samples. The samples should be at an appropriate Hct level that is within the target range of the patient group. It is important to ensure that the pooled sample is analysed before being enriched with the analyte(s) of interest to ensure that the endogenous concentration of the analyte is known and that there is no interference from endogenous analytes or drugs. It is recognised that overall the non-matrix spike volume should be <5%, otherwise the blood sample matrix is compromised, as this can affect the Hct level or even cause haemolysis and therefore sample homogeneity. The intra- and inter-assay precision, from the replicate analyses, should be ≤15%. The bias from the enrichment concentration values should be <±15% [52]. However, the performance of DBS assays used to monitor patients may need to be more stringent, especially when patient results are compared to consensus target treatment ranges [5].

### 3.7. DBS External Quality Assurance (EQA) Schemes

It is recommended that all laboratories should participate in EQA schemes or sample exchange when an EQA scheme is not available, as it is an essential tool to ensure that its performance is comparable to that of other laboratories. The UK National External Quality Assessment Service (NEQAS) and the CDC schemes for DBS analytes have been available for many years. However, these schemes are intended to assess the performance of newborn screening programs. In order to address this issue, the European Research Network for evaluation and improvement of screening, Diagnosis and treatment of Inherited disorders of Metabolism (ERNDIM) introduced a DBS scheme in 2017 targeted towards the monitoring of IMD patients receiving treatment. The following analytes are included in the EQA scheme: Allo-isoleucine, C0, Homocysteine, Isoleucine, Leucine, Methionine, NTBC, Phenylalanine, SUAC, Tyrosine and Valine. 

The pilot scheme was operational in 2017 and 2018 and supplied 79 laboratories with four DBS specimens covering the range of concentrations that would be observed in patients. The results from this EQA pilot scheme indicate that there are significant problems with assay calibration resulting in an unwanted level of inter-laboratory variation and an inherent results bias in some laboratories for the various analytes. As of 2020, there are 87 participants in the ERNDIM DBS EQA scheme.

## 4. Post-Analytical Factors

Several factors can affect the final test result: transcription errors, understanding of the assay performance (precision and accuracy), clinical interpretation and turnaround times. However, a major post-analytical issue is related to the difference between analyte concentration in the DBS versus plasma specimens and the comparison of the result to consensus patient target treatment ranges.

### 4.1. Translation of DBS Results to Plasma Concentrations

Plasma/serum are often the most common matrices used for diagnosis and monitoring of IMD patients, most notably for amino acids. The amino acids phenylalanine, tyrosine, leucine, isoleucine and allo-isoleucine were shown to be highly correlated but negatively biased relative to plasma concentrations [4]. Many of the analytes routinely measured in DBS have been shown to be lower than in paired plasma samples (Table 1). The observed differences between DBS and plasma concentrations are due to several factors: distribution of the analyte between the plasma and erythrocytes, extraction efficiency from DBS, sample preparation, derivatisation of sample and methodological test biases. NTBC is also reported to be significantly lower in DBS vs. plasma concentrations [10,11]. C0 concentrations are reported to be ~30–40% higher in plasma compared to DBS specimens [15,16]. DBS MMA concentrations have been reported to be ~5–6 times lower than paired plasma specimens [9]. Theoretically, MMA concentrations should be ~50% of those in plasma, if the erythrocytes contain no MMA. This 5–6 fold difference is most likely due to standardisation/calibration calculation issues. Lyso-Gb3 concentrations in plasma and DBS specimens were shown to be comparable in one study [17], and ~50% lower in DBS vs. plasma specimens in another study [18]. Based upon these observations, it is therefore imperative that clinical laboratories undertake appropriate validation studies to assess the difference in analyte concentration between plasma and DBS specimens.

### 4.2. Comparison of Results to Target Treatment Ranges

Guidelines for the diagnosis and management of patients with PKU, MSUD and homocystinuria have been published [53,54,55]. One of the key recommendations is the monitoring of the protein-restricted diet, using metabolite concentrations interpreted against consensus target treatment ranges to prevent the adverse outcomes observed in these conditions. It is recommended that patients with tyrosinaemia type 1 are treated with NTBC and be monitored using DBS SUAC, as it is a sensitive indicator for suboptimal NTBC treatment [56]. However, the limit of detection of the assay and variable extraction efficiencies may affect the ability to detect small increases in SUAC. The target treatment range for DBS NTBC is 30 to 70 µmol/L [57]. However, these target ranges are impeded by the lack of assay standardisation and variable extraction efficiencies. DBS 17-OHP is a useful marker to monitor response to therapy in patients with CAH, as multiple specimens can be collected over a 24 h period [58]. 

The reporting of inaccurate monitoring results could have profound effects in that patients may be falsely reassured with lower results, where laboratories have a negative bias for DBS analytes and conversely, those laboratories with a positive bias providing falsely elevated results, which may prompt a stricter dietary/treatment regimen, which may lead to non-compliance issues. Therefore, clinical laboratories should be aware of test bias when utilising the recommended target treatment ranges. To provide comparable results for patient monitoring, a calibration factor could be used to report DBS results as plasma equivalents to ensure meaningful comparison of results to the recommended target treatment ranges. This is preferable to reporting patient results against different target treatment ranges as this may cause confusion for both the patient and the clinician. Particularly in conditions where both specimen types are used, e.g., plasma amino acids and DBS phenylalanine in patients with PKU [5]. It is essential that a rigorous evaluation of the bias between plasma and DBS analyte results is undertaken in laboratories to derive a calibration factor in order to report DBS results as plasma equivalents (ideally on an individual patient basis), thereby ensuring meaningful comparison of patient results to the recommended target treatment ranges [5].

## 5. Conclusions 

The use of DBS specimens for IMD patient monitoring can offer practical, clinical and financial advantages due to the convenience of sample collection, transport, storage and bio-safety when compared to conventional liquid blood collection methods. With the development of sensitive UPLC–MS/MS instruments, microfluidics, and immuno-assay systems, the monitoring of patients with IMDs could therefore rely on the use of DBS specimens and pave the way for further assay development. Given that UPLC–MS/MS methods are superior in terms of specificity and precision and have a comparable analysis time relative to FIA–MS/MS, it is recommended that laboratories move to implementing such methods. It is imperative that the technologies used for patient monitoring are rapid and reproducible to enable high sample throughput to ensure shorter turnaround times for monitoring results, as this can optimize outcomes for patients. 

Although the technique of DBS collection and analysis has been used in NBS since the 1960s, there has been a recent explosion of interest in the use of DBS specimens in the pharmaceutical industry for pre-clinical pharmacokinetic studies and for population-based studies [59]. Numerous studies have now been published assessing the impact of variables that can affect the quantification of analytes in DBS specimens. This increased knowledge has resulted in recommendations on the validation of analytical methods for DBS specimens by the European Bioanalysis Forum [51]. This recommendation paper identifies a large number of factors that need to be assessed in the development and validation of DBS assays and is an excellent resource to guide clinical laboratories.

The majority of published studies applicable to the monitoring of IMD patients have utilised the PerkinElmer 226 or Whatman 903 blood collection devices—both of which are FDA approved. Other devices which are commercially available have different properties (i.e., paper thickness and density) which influence the adsorption and dispersion of blood. In addition, filter paper materials are being developed to enhance analyte stability and extraction by chemical treatment of the paper. Many of these filter paper devices have not been evaluated against the CLSI NBS01 criteria and therefore the consistency of the paper production across lots has not been evaluated. It is therefore important to evaluate the lot-to-lot variance of the filter paper device as part of the method evaluation when using non-CLSI-verified materials for the monitoring of patients. 

Temperature and humidity have significant effects on the DBS analytes that are used to monitor patients with IMDs. It is recommended that those laboratories offering a clinical monitoring service for the various DBS analytes should undertake rigorous studies to assess the effects of temperature changes and humidity on analyte concentrations in DBS specimens. This is particularly important when the specimen might be exposed to uncontrolled environmental conditions during collection, transportation and storage. 

DBS specimen volume/size, quality and Hct can all affect analytical results significantly and therefore have major implications when monitoring patients with IMDs. Hct varies significantly with age and gender and can also vary significantly during periods of decompensation or as a result of hydration status (i.e., patient may be de-hydrated or be receiving intravenous fluid therapy) and disease states (e.g., liver disease). It is important to recognise that patient results may vary significantly and that analysis of DBS specimens may not be the first choice of specimen in certain situations. Assessing the diameter of the DBS may be used as a tool to identify samples that may give unacceptable analytical biases for analyte concentrations [39]. Laboratories should have standardised criteria to ensure that all laboratories are accepting and rejecting samples of the same size and quality [25,39]. Only appropriately sized samples should be accepted for DBS assays, thereby ensuring accurate monitoring results. Overcoming DBS specimen volume, quality, and Hct issues could potentially be achieved by the use of blood collection devices that collect defined volumes of liquid blood for sampling and such devices should be evaluated in order to improve the biochemical monitoring of patients with IMDs. Until then, patients and parents/carers should receive regular training on blood collection techniques to ensure that more accurate and less variable results are obtained in order to achieve optimal dietary or therapeutic control, thereby reducing adverse outcomes associated with the various disorders.

The results from the ERNDIM DBS EQA pilot scheme in 2017 and 2018 indicate that there are significant problems with assay calibration, resulting in an unwanted level of inter-laboratory variation and an inherent results bias in some laboratories for the various analytes. Although aqueous CRM is available for some of the metabolites used to monitor patients with IMDs, currently there are no commercially available CRMs for these metabolites in DBS specimens. An international effort between professional societies, expert scientific advisory groups, patient advocacy groups and organizations that have the expertise and capabilities to produce DBS CRM material is required, in order to standardize DBS tests. Reporting inaccurate monitoring results could have profound effects. Patients may be falsely reassured with lower results, where laboratories have a negative bias for DBS analytes. Conversely, those laboratories with a positive bias providing falsely elevated results may prompt a stricter treatment regimen, which may lead to non-compliance issues. Therefore, with such large and variable biases for DBS analyte results being observed between different laboratories, it is evident that consideration should be given to test bias when utilising consensus target treatment ranges. Clinicians should take into consideration the effect of the test variability and bias (i.e., the total error of the test), DBS size and quality in order to prevent over-interpretation of changes in analyte concentrations, thereby preventing false reassurances as to optimal therapy.

## Figures and Tables

**Figure 1 IJNS-06-00026-f001:**
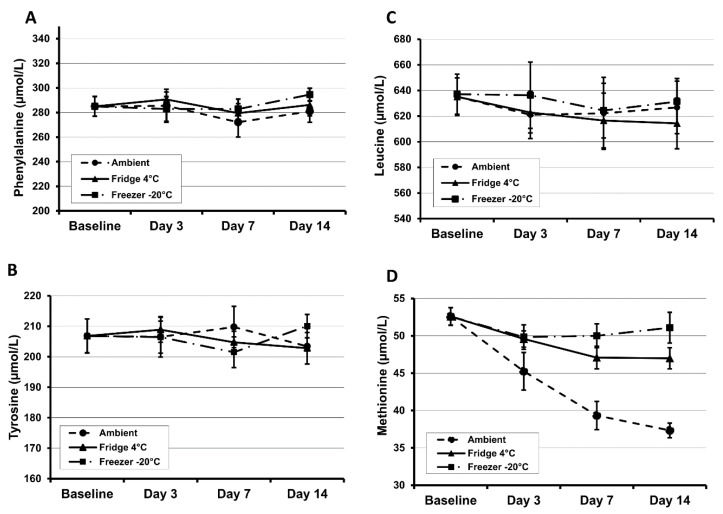
Concentration changes of phenylalanine (**A**), tyrosine (**B**), leucine (**C**) and methionine (**D**) over time in DBS samples (PerkinElmer 226 cards) at various storage temperatures. Results are shown as the mean ± SD; *n* = 12 replicates in each experiment.

**Figure 2 IJNS-06-00026-f002:**
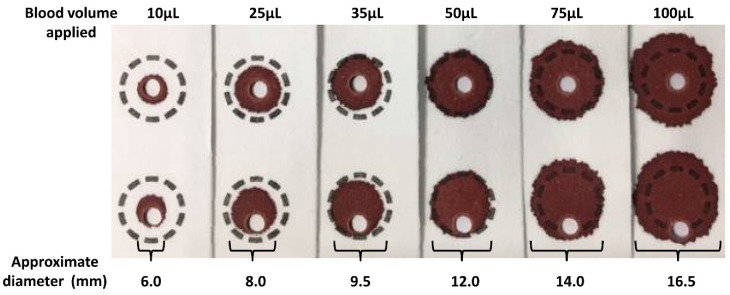
Relationship between volumes of blood applied to the filter paper collection devices and the measured dried blood spot (DBS) diameter. Results shown are the mean of the diameter measurements (results from [25,39]). Sub-punches (3.2 mm) were taken from central (first row) and peripheral locations (second row).

**Figure 3 IJNS-06-00026-f003:**
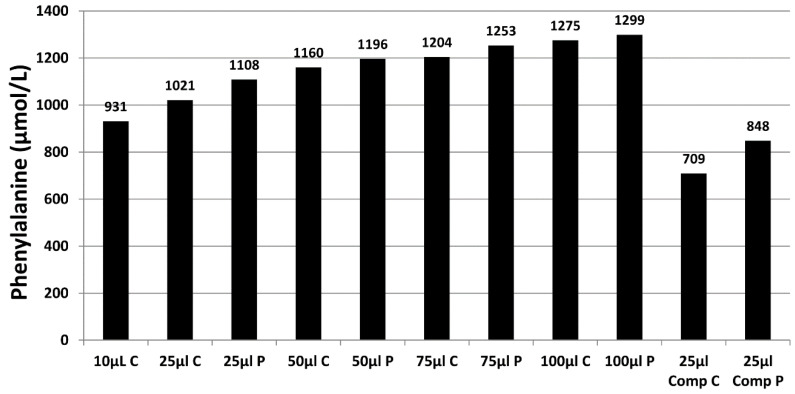
Effect of sample volume, punch location and compression of the sample on DBS phenylalanine concentrations in a patient with phenylketonuria (PKU) (results are a mean of *n* = 3). C, central punch; P, peripheral punch; Comp, compressed.

**Figure 4 IJNS-06-00026-f004:**
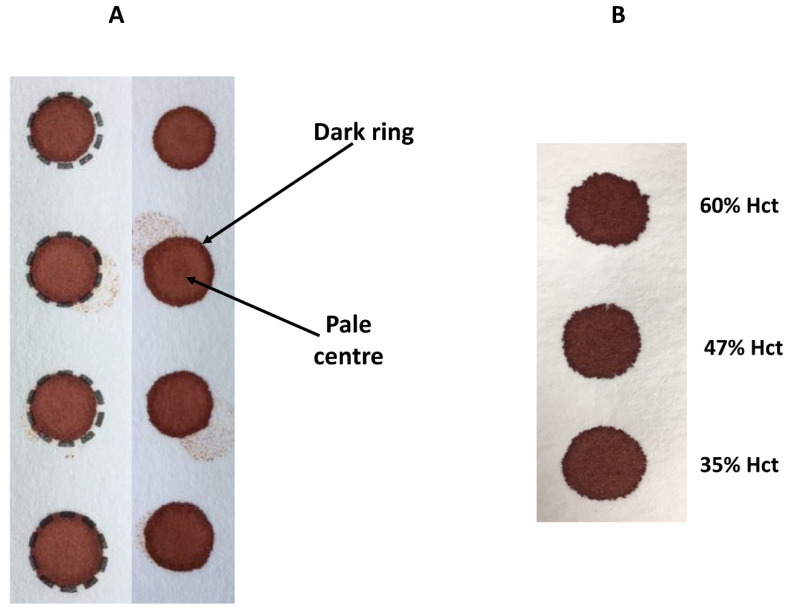
Effect of compression on DBS appearance (**A**) and varying haematocrit (Hct) on DBS appearance (**B**).

**Table 1 IJNS-06-00026-t001:** List of analytes measured in dried blood spot (DBS) specimens for the monitoring of patients with various inherited metabolic disorders (IMDs).

Analyte	IMD	Laboratory Instrumentation	Diameter of DBS Used for Analysis	Assay Performance Characteristics	References
Phenylalanine	PKU	HPLC, UPLC, FIA–MS/MS, LC–MS/MS	2 × 6 mm 3.2 mm	UPLC (derivatised) 0.5–197 µmol/L, CVs < 10%, DBS ~36% lower vs. plasma Various studies—DBS 15–28% lower vs. plasma	[4,5]
Tyrosine	PKU, Tyro I, II and III, AKU	HPLC, UPLC, FIA–MS/MS, LC–MS/MS	2 × 6 mm 3.2 mm	UPLC (derivatised) 0.5–197 µmol/L, CVs < 10% DBS ~38% lower vs. plasma	[4]
Methionine	HCU	HPLC, UPLC, FIA–MS/MS, LC–MS/MS	3.2 mm	0.5–197 µmol/L, CVs < 10% DBS 40% lower vs. plasma	[6]
Homocysteine	Homocystinuria	LC–MS/MS, HPLC	6 mm, 3.2 mm	1–100 µmol/L, CVs < 10%, DBS 40–50% lower vs. plasma	[6,7,8]
Leucine	MSUD	HPLC, UPLC, FIA–MS/MS, LC–MS/MS	3.2 mm 2 × 6 mm	UPLC (derivatised) 0.5–197 µmol/L, CVs < 10% DBS ~ 40% lower vs. plasma	[4]
MMA	Methylmalonic aciduria	LC–MS/MS	8 mm	10–10,000 nmol/L, recovery 95%, CVs < 5% DBS 5–6× lower vs. plasma	[9]
NTBC	Tyro1, AKU	HPLC, LC–MS/MS	3.2 mm 3.2 mm	0.1–100 µmol/L, recovery 73%, CVs < 10%, DBS 1.56 × lower vs. plasma 0.05–50 µmol/L, recovery 100%, CVs < 10%, plasma 2.4 × higher vs. DBS	[10,11]
SUAC	Tyro1	LC–MS/MS	3.2 mm	0.1–100 µmol/L, recovery > 99%, CVs < 10% 3.0–250 µmol/L, recovery > 99%, CVs < 6%	[12,13,14]
CO	FAOs and OAs	FIA–MS/MS, LC–MS/MS	3.2 mm	0–150 µmol/L, CVs < 10% DBS 30–40% lower vs. plasma	[15,16]
Lyso-Gb3	Fabry	LC–MS/MS	3.2 mm 3.2 mm	0.45–197 nmol/L, recovery—NR, CVs < 10%, DBS and plasma results comparable 0.0–120 ng/mL, DBS ~ 50% lower vs. plasma	[17,18]
17-OHP	CAH	LC–MS/MS	3.2 mm	10–200 nmol/L, recovery 90–110%, CVs < 10%, DBS ~ 35–50% lower vs. plasma	[19,20,21]

PKU, phenylketonuria; Tyro, tyrosinaemia; AKU, alkaptonuria; MSUD, Maple Syrup Urine Disease; MMA, methylmalonic acid; NTBC, nitisinone; SUAC, succinylacetone; CO, free carnitine; FAODs, Fatty Acid Oxidation Disorders; OAs, Organic Acidurias; Lyso-Gb3, globotriaosylsphingosine; 17-OHP, 17-hydroxyprogesterone; CAH, Congenital Adrenal Hyperplasia; NR, not reported.

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
