# Peer review of "Use of Dried Blood Spot Specimens to Monitor Patients with Inherited Metabolic Disorders"

_2409-515X, 2020, doi:10.3390/ijns6020026_

Round 1

Reviewer 1 Report

This is an excellent review paper on the use of dried blood spots to monitor treatment of various metabolic disorders. The authors describe in detail the approved filter papers, the method of collecting the blood from a heel or finger prick, the evaluation of the blood spot by the assay laboratory, and the potential loss of various analytes secondary to storage conditions. There is a useful table organized by analyte with references to assay methods, instrumentation, recovery of analyte from the DBS, coefficients of variation and a comparison of recovery of the analyte from DBS compared with plasma.

The paper will be a useful reference for metabolic laboratories using the dried blood spot for monitoring treatment of inherited metabolic disorders that are identified by newborn screening.

Author Response

Reviewer #1:

This is an excellent review paper on the use of dried blood spots to monitor treatment of various metabolic disorders. The authors describe in detail the approved filter papers, the method of collecting the blood from a heel or finger prick, the evaluation of the blood spot by the assay laboratory, and the potential loss of various analytes secondary to storage conditions. There is a useful table organized by analyte with references to assay methods, instrumentation, recovery of analyte from the DBS, coefficients of variation and a comparison of recovery of the analyte from DBS compared with plasma.

The paper will be a useful reference for metabolic laboratories using the dried blood spot for monitoring treatment of inherited metabolic disorders that are identified by newborn screening.

 No response required.

 Reviewer 2 Report

    The authors have properly reviewed the information about quality of bloodspot collection devices, methods of blood sampling, application of blood on bloodspot collection devices, punching sites of DBS, and metabolite instability during storage, but have not given sufficient comment on “significant problems with assay calibration resulting in an unwanted level of inter-laboratory variation” in the ERNDIM trial. It seems important for the authors to stress in Abstract that many issues still remain in case metabolite values in DBS obtained by FIA-MS/MS analysis are applied to evaluate the patient’s condition under medical treatment.

Individual specific comments are as follows;

5.0 Conclusion

   As mentioned in Introduction, it seems that FIA-MS/MS is not a good measure for clinical use because of lack of traceability [A].

   Most importantly, in FIA-MS/MS analysis, matrix and target metabolites are not separated during electrospray-ionization process, and the consistency of ionization may be impaired. This kind of inconsistency seems to be the major cause of wide inter-instrument and inter-laboratory variation of the measured values. On the other hand, in LC-MS/MS analysis, the effect by matrix-associated ion suppression is practically reduced, although phospholipids may still interfere the quantitation of the target metabolite by co-elution in chromatography [56, B, C].

  Additional limitation is the issue about internal standards. Several IS-Kit suppliers use d8-valine, d5- or 13C6-phenylalanine, d4-citllurine, d9-carnitine, d9-isovalerylcarnitine, and d9-glutarylcarnitine as internal standards. It is assumed that the SRM (MRM) conditions for both native and stable-isotope labeled metabolites mentioned above are rather difficult to be optimized properly according to the manner designated by instrument manufacturers.

  We have recently done FIA-MS/MS study concerning the correlation between amino acids or acylcarnitines and the corresponding labeled internal standards using aqueous calibrator and IS-kits supplied by Perkin Elmer, CIL and SEAMENS in order to clarify the influence of the stable-isotope labeling (d9-, d8, d3-, etc) together with the ability of the MS/MS instruments during routine laboratory work in 36 screening laboratories in Japan. Although this kind of FIA-MS/MS analysis is thought to be hardly influenced by matrix effects, the ratios of the ion intensities of native metabolites to those of the corresponding labeled ones (IS) did not converge to the theoretical value calculated using the amounts of native metabolite and IS which were shown in the attached documents, but ranged widely. Our results suggested that labeling nature of IS may affect MS/MS measurement and the optimization of SRM (MRM) conditions may be difficult, partly because the daily work to keep the performance of the instrument against pollution by sample injections may not be easy.

 Regarding traceability, purity and stability for such labeled acylcarnitines as d9-glutarylcarnitine and d3-3-hydroxxyisovalerylcarnitine seemed difficult to be guaranteed by IS-kit suppliers. This may affect the trueness value assignment of the measured values in MS/MS analysis [A].

  Considering the issues mentioned above, I would like to ask the authors to comment on my suggestion that LC-MS/MS analysis should be recommended for clinical laboratories which provide values for clinical use, that is, monitoring of IMDs patients.

3.5. Assay Calibration

    I agree with the authors in terms of “the lack of commercially available certified reference material (CRM) for the various analytes on which to standardise laboratory tests”. On the other hands, it is known that Perkin  Elmer is providing DBS calibrators in the IS kit approved by FDA, although I am not sure if the quality could meet to the conditions which the authors gave in this article. If the values of the metabolites in DBS calibrators were certified by LC-MS/MS, DBS calibrators provided by IS kit suppliers could be CRM. I would like to ask the authors to comment on DBS calibrators in the IS kit supplied by kit makers. I can suggest an article [D] about a candidate of LC-MS/MS method applicable to this kind of project.

3.6. DBS Internal Quality Control

    Discussion in this section seems confusing, since the methods for MS/MS analysis were not stated; FIA-MS/MS or LC-MS/MS?

    It seems important to discuss quality control under the assumption that DBS analysis must be performed by LC-MS/MS. Even in FIA-MS/MS analysis of DBS, the suggestion that the intra- and inter-assay precision should be ≤15 % may be less strict.

3.7. DBS External Quality Assurance (EQA) Schemes

    It is stated that ERNDIM introduced a DBS scheme in 2017 targeted towards the monitoring of IMD patients receiving treatment. Is it correct that it was done by FIA-MS/MS?  I am afraid that DBS measurement by FIA-MS/MS may result in confusing situations. I suggest that they should move to LC-MS/MS project or they use the special DBS calibrator for FIA-MS/MS after the values of DBS be guaranteed by the authorized LC-MS/MS according to the manner of traceability.

4.1 Translation of DBS results to plasma concentrations

    Based on the discussion mentioned above, the authors should precisely describe the methods in Table 1 in order to clarify which values were obtained by FIA-MS/MS or LC-MS/MS.

    I cannot access the article #40 and #52 through PubMed website, and so, I cannot properly comment on the data about the difference of phenylalanine levels in DBS and those in plasma. My experience is that the levels of amino acids in DBS and those in plasm did not match at all during acute illnesses causing ketoacidosis, while both levels were similar during stable conditions. Although transport for aromatic amino acids across erythrocyte membrane is reported to be almost passive [E], it is practical to suppose that there is lag time until amino acid levels in DBS and in plasma reach equilibrium. Probably, plasma levels of phenylalanine in PKU patients fluctuate according to diet intake, and blood-sampling time may significantly affect the difference between the levels in DBS and those in plasma.

    Free carnitine (C0) levels in DBS vs plasma may be affected partly by such sample preparation as derivatization [42], and partly by the nature of erythrocytes, the remnants of erythroblasts, containing C0 related to membrane transport and disease-specific metabolic process. Thus, the authors should comment on the other factors than analytical process in addition to “DBS 30-40% lower vs plasma” in assay performance characteristics.

    The conclusion of six-times lower methylmalonic acid (MMA) levels in DBS than those in plasm in the article [56] seems confusing. Theoretically, MMA levels in DBS should be around the half of those in plasma if erythrocytes contain no MMA. This method by LC-MS/MS may have some errors concerning traceability, but I could not find the calibration curve and equation in MMA quantitation in this article. Otherwise, the results in 17-OHP measurement by LC-MS/MS [57-59] seems reasonable. 17-OHP analysis by LC-MS/MS was described as a good example in the article of traceability [A]. I would like to ask the authors to add the comments about the results of the articles in the Table 1.

I have found an error of spelling "asses" in line 67, which could be "assess".

[References]

A. Koumantakis G: Traceability of measurement results. Clin Biochem Rev. 29 Suppl (i) August, 2008.

B. An G, Bach T, Abdallah I, Nalbant D: Aspects of matrix and analyte effects in clinical pharmacokinetic sample analyses using LC-ESI/MS/MS - Two case examples. J Pharm Biomed Anal. 2020 Jan 30;183:113135.

C. Trivedi V, Shah PA, Shrivastav PS, Sanyal M: Optimization of chromatography to overcome matrix effect for reliable estimation of four small molecular drugs from biological fluids using LC-MS/MS. Biomed Chromatogr. 2020 Mar;34(3): e4777.

D. Miller JH 4th, Poston PA, Karnes HT: A quantitative method for acylcarnitines and amino acids using high resolution chromatography and tandem mass spectrometry in newborn screening dried blood spot analysis. J Chromatogr B Analyt Technol Biomed Life Sci. 15;903:142-9, 2012.

E. Hagenfeldt L, Arvidsson A: The distribution of amino acids between plasma and erythrocytes. Clin Chim Acta. 100(2):133-41, 1980.

Author Response

I will address the points in the order as raised by the reviewers. The responses are in bold after each of the points raised by the reviewers.

Reviewer #2:

The authors have properly reviewed the information about quality of bloodspot collection devices, methods of blood sampling, application of blood on bloodspot collection devices, punching sites of DBS, and metabolite instability during storage, but have not given sufficient comment on “significant problems with assay calibration resulting in an unwanted level of inter-laboratory variation” in the ERNDIM trial. It seems important for the authors to stress in Abstract that many issues still remain in case metabolite values in DBS obtained by FIA-MS/MS analysis are applied to evaluate the patient’s condition under medical treatment.

As the abstract states “In this review we discuss the pre-analytical, analytical and post-analytical variables that may affect the final test result obtained using DBS specimens used for monitoring of patients with an IMD.” We have included a further statement in the Conclusion please refer to lines 451- 452.

Given that UPLC-MS/MS methods are superior in terms of specificity, precision and have comparable analysis time relative to FIA-MS/MS it is recommended that laboratories move to implementing such methods.

5.0 Conclusion

   As mentioned in Introduction, it seems that FIA-MS/MS is not a good measure for clinical use because of lack of traceability [A].

   Most importantly, in FIA-MS/MS analysis, matrix and target metabolites are not separated during electrospray-ionization process, and the consistency of ionization may be impaired. This kind of inconsistency seems to be the major cause of wide inter-instrument and inter-laboratory variation of the measured values. On the other hand, in LC-MS/MS analysis, the effect by matrix-associated ion suppression is practically reduced, although phospholipids may still interfere the quantitation of the target metabolite by co-elution in chromatography [56, B, C].

We have amended this section on lines 46-51 – See below:

However, it should be recognised that FIA-MS/MS methods lack specificity. The absence of chromatographic separation means that specificity is achieved solely by the use of selective reaction monitoring (SRM). Consequently any isobaric compound with a common daughter ion has the potential to interfere. A further disadvantage is that the analyte(s) of interest is not separated from the sample matrix, which can result in non-specific interferences from phospholipids and salts.

Additional limitation is the issue about internal standards. Several IS-Kit suppliers use d8-valine, d5- or 13C6-phenylalanine, d4-citllurine, d9-carnitine, d9-isovalerylcarnitine, and d9-glutarylcarnitine as internal standards. It is assumed that the SRM (MRM) conditions for both native and stable-isotope labeled metabolites mentioned above are rather difficult to be optimized properly according to the manner designated by instrument manufacturers.

We have recently done FIA-MS/MS study concerning the correlation between amino acids or acylcarnitines and the corresponding labeled internal standards using aqueous calibrator and IS-kits supplied by Perkin Elmer, CIL and SEAMENS in order to clarify the influence of the stable-isotope labeling (d9-, d8, d3-, etc) together with the ability of the MS/MS instruments during routine laboratory work in 36 screening laboratories in Japan. Although this kind of FIA-MS/MS analysis is thought to be hardly influenced by matrix effects, the ratios of the ion intensities of native metabolites to those of the corresponding labeled ones (IS) did not converge to the theoretical value calculated using the amounts of native metabolite and IS which were shown in the attached documents, but ranged widely. Our results suggested that labeling nature of IS may affect MS/MS measurement and the optimization of SRM (MRM) conditions may be difficult, partly because the daily work to keep the performance of the instrument against pollution by sample injections may not be easy.

 Regarding traceability, purity and stability for such labeled acylcarnitines as d9-glutarylcarnitine and d3-3-hydroxxyisovalerylcarnitine seemed difficult to be guaranteed by IS-kit suppliers. This may affect the trueness value assignment of the measured values in MS/MS analysis [A].

Considering the issues mentioned above, I would like to ask the authors to comment on my suggestion that LC-MS/MS analysis should be recommended for clinical laboratories which provide values for clinical use, that is, monitoring of IMDs patients.

The reviewer raises a valid point in that we should emphasise that the clinical laboratories should move from using FIA-MS/MS to UPLC-MS/MS assays to monitor patients with IMDs.

We have included a further statement in the Conclusion please refer to lines 451- 453.

Given that UPLC-MS/MS methods are superior in terms of specificity, precision and have comparable analysis time relative to FIA-MS/MS it is recommended that laboratories move to implementing such methods.

3.5. Assay Calibration

   I agree with the authors in terms of “the lack of commercially available certified reference material (CRM) for the various analytes on which to standardise laboratory tests”. On the other hands, it is known that Perkin Elmer is providing DBS calibrators in the IS kit approved by FDA, although I am not sure if the quality could meet to the conditions which the authors gave in this article. If the values of the metabolites in DBS calibrators were certified by LC-MS/MS, DBS calibrators provided by IS kit suppliers could be CRM. I would like to ask the authors to comment on DBS calibrators in the IS kit supplied by kit makers. I can suggest an article [D] about a candidate of LC-MS/MS method applicable to this kind of project.

There are a number of sources of internal standards and commercially available kits. The proposal that a single LC-MS/MS method used to certify DBS calibrators from a commercially available kit does not constitute a CRM. CRM material has to be produced by expert independent organisations e.g. NIST that conform to the relevant ISO regulations and have defined traceability etc.

 The point we are making is that due to the lack of matrix matched CRM we observe large inter-lab variations.

 We have included an additional section in the discussion – please refer to lines 499-503. See below:

 Although aqueous CRM is available for some of the metabolites used to monitor patients with IMDs, currently there are no commercially available CRM’s for these metabolites in DBS specimens. An international effort between professional societies, expert scientific advisory groups, patient advocacy groups and organizations that have the expertise and capabilities to produce DBS CRM material is required, in order to standardize DBS tests.

3.6. DBS Internal Quality Control

   Discussion in this section seems confusing, since the methods for MS/MS analysis were not stated; FIA-MS/MS or LC-MS/MS?

The reviewers comment does not make sense as the preparation of QC material is independent of whether or not it is analysed by FIA-MS/MS or UPLC-MS/MS. The point we are making is that the preparation of calibrators and QC material using different in Hct and size (volume) etc will have an impact on the results whatever method is used!

    It seems important to discuss quality control under the assumption that DBS analysis must be performed by LC-MS/MS. Even in FIA-MS/MS analysis of DBS, the suggestion that the intra- and inter-assay precision should be ≤15 % may be less strict.

Unfortunately, there is little guidance on minimum performance criteria for DBS assays. At present guidance from the FDA is rather generic. We have therefore inserted a comment on lines 375-377 - However, the performance of DBS assays used to monitor patients may need to be more stringent, especially when patient results are compared to consensus target treatment ranges.

 3.7. DBS External Quality Assurance (EQA) Schemes

   It is stated that ERNDIM introduced a DBS scheme in 2017 targeted towards the monitoring of IMD patients receiving treatment. Is it correct that it was done by FIA-MS/MS? I am afraid that DBS measurement by FIA-MS/MS may result in confusing situations. I suggest that they should move to LC-MS/MS project or they use the special DBS calibrator for FIA-MS/MS after the values of DBS be guaranteed by the authorized LC-MS/MS according to the manner of traceability.

We are not sure what the reviewer is asking here?

 For the ERNDIM EQA scheme - the laboratories use a variety of methods. It is my understanding that the results of this pilot study are to be written up and published at a later date. But those metabolic laboratories that participate in the scheme recognise that the lack of matrix matched certified material affects ALL methods.

4.1 Translation of DBS results to plasma concentrations

   Based on the discussion mentioned above, the authors should precisely describe the methods in Table 1 in order to clarify which values were obtained by FIA-MS/MS or LC-MS/MS.

Table 1 was included to provide the reader with a few key references for the analytes that are most commonly analysed to monitor patients with IMDs. It is therefore not an exhaustive list and if the reader requires further information they can then refer to the reference.

    I cannot access the article #40 and #52 through PubMed website, and so, I cannot properly comment on the data about the difference of phenylalanine levels in DBS and those in plasma. My experience is that the levels of amino acids in DBS and those in plasm did not match at all during acute illnesses causing ketoacidosis, while both levels were similar during stable conditions. Although transport for aromatic amino acids across erythrocyte membrane is reported to be almost passive [E], it is practical to suppose that there is lag time until amino acid levels in DBS and in plasma reach equilibrium. Probably, plasma levels of phenylalanine in PKU patients fluctuate according to diet intake, and blood-sampling time may significantly affect the difference between the levels in DBS and those in plasma.

Reference 52 is an article that reviews in depth the performance of lab assays used to quantitate phenylalanine. It highlights that 7 peer reviewed studies have demonstrated that DBS phenylalanine concentrations are 15 to 28% lower in DBS (analysed by FIA-MS/MS) vs paired plasma samples (analysed by IEC). The difference between DBS and plasma are more variable when different methods are used to measure the DBS and plasma samples.

    Free carnitine (C0) levels in DBS vs plasma may be affected partly by such sample preparation as derivatization [42], and partly by the nature of erythrocytes, the remnants of erythroblasts, containing C0 related to membrane transport and disease-specific metabolic process. Thus, the authors should comment on the other factors than analytical process in addition to “DBS 30-40% lower vs plasma” in assay performance characteristics.

Both of studies that we quote [42, 43] that assessed the difference between plasma and DBS free carnitine both used a derivatisation method (butanolic HCL). Therefore, both sample types were analysed using the same methodology.

 In section 4.1 we have made the point on line 403 – “The observed differences between DBS and plasma concentrations are due to several factors; distribution of the analyte between the plasma and erythrocytes, extraction efficiency from DBS and methodological test biases.”

 We have modified this comment to include… sample preparation, derivatisation of sample (see line 407).

  The conclusion of six-times lower methylmalonic acid (MMA) levels in DBS than those in plasm in the article [56] seems confusing. Theoretically, MMA levels in DBS should be around the half of those in plasma if erythrocytes contain no MMA. This method by LC-MS/MS may have some errors concerning traceability, but I could not find the calibration curve and equation in MMA quantitation in this article. Otherwise, the results in 17-OHP measurement by LC-MS/MS [57-59] seems reasonable. 17-OHP analysis by LC-MS/MS was described as a good example in the article of traceability [A]. I would like to ask the authors to add the comments about the results of the articles in the Table 1.

We agree with the reviewer in that the difference between DBS and plasma MMA in paired samples does not make sense (hence the inclusion of the difference in concentrations between DBS and plasma in the table) and reflect the fact that the method used has a significant issue in terms of standardisation etc. We have inserted a sentence in section 4.1 lines 410-413 – “DBS MMA concentrations have been reported to be ~5-6 times lower than paired plasma specimens [18]. Theoretically, MMA concentrations should be ~50% of those in plasma, if the erythrocytes contain no MMA. The 5-6 fold difference is most likely due to standardisation / calibration calculation issues”.

 Please note that we identified an error with reference 56 – This was quoted earlier as reference 18. We have now amended this error.

I have found an error of spelling "asses" in line 67, which could be "assess".

This typo has been corrected.

Reviewer 3 Report

General comments and appreciation

The present manuscript presents a review on the use of DBS for patient monitoring. The use of DBS as sample for analytical procedures other than NBS has been subject of other papers, but focusing mainly in drug monitoring/screening. The manuscript is an original review and scientifically valuable for those working in the field. It is well written and organized focusing the different aspects of the subject. In my opinion the paper should be accepted for publication; nevertheless some minor revisions are needed:

  1. Lines 21 -23
    1. This is true, but not indispensable for the use of DBS for patient monitoring. I suggest a reformulation of the sentence to make it clearer.
  2. Lines 46 – 47
    1. Although theoretically chromatographic methods are associated with superior specificity, the specificity of FIA-MS/MS is related to the analyte. For some it is specific, for others it isn’t. Please consider reformulating the sentence.
  3. Lines 48 -50
    1. It is said that specificity in FIA-MS/MS is achieved using SRM, and them it is said that isobaric compounds with a common daughter ion may interfere in quantification. This seams a bit contradictory. On the other hand, SRM is also known for being a more sensitive approach than PS, NL, or other full scan modes. Please consider readjusting the sentence.
  4. Line 81
    1. In the section DBS specimen collection devices, please consider including considerations on how to store the filter paper cards until they are used as well on the expiration date.
  5. Line 163
    1. It is said that rapid drying improves stability. Please consider adding that it should not be dried on heat sources.
  6. Line 223
    1. It is said that a 3.2 mm punch yields for 3.0μL of sample. This is influenced by many factors, as very well said during the text, playing haematocrit a key role. Please consider indicating a volume interval or referring the value of 3.0μL as an average.
  7. Line 228
    1. Please consider including the key importance of the signal/noise ratio.
  8. Line 311 – 314
    1. It is said that SUAC requires hydrocloridric acid for elution and ethylacetate for extraction and then that SUAC can be extracted using acetonitrile-water-formic acid containing hydrazine. This seams contradictory. Please consider reformulate the sentence.
  9. Line 343
    1. There are CRM for some analytes. Please reformulate.
  10. Line 351
    1. In this section please consider adding one of the problems when preparing QC samples, that is haemolysis caused by the addition of non-matrix volumes.
  11. Line 404
    1. It is said that patients should be monitoring using DBS. This is not stated in ref 49. Although with advantages, maybe “should” is a bit too much. Please consider changing it.
  12.  All document
    1. As it is said in the text, one of the main factors influencing the quantification of analytes in DBS is haematocrit. In NBS applications is assumed that it is more or less constant in the first few days of life. But in older infants and adults this is not necessarily true, namely in decompensation/sick periods. So it may be worthy to stress that when using DBS for patient follow-up, the results should be affected by the values of average aged match haematocrit and that DBS are not the best choice for many of the hospitalized patients (under intravenous fluid therapy, decompensation periods, etc). In my experience, the idea that DBS are not the best option in every situation is difficult to assimilate for many heath professionals.

Author Response

Reviewer #3:

 The present manuscript presents a review on the use of DBS for patient monitoring. The use of DBS as sample for analytical procedures other than NBS has been subject of other papers, but focusing mainly in drug monitoring/screening. The manuscript is an original review and scientifically valuable for those working in the field. It is well written and organized focusing the different aspects of the subject. In my opinion the paper should be accepted for publication; nevertheless some minor revisions are needed:

  1. Lines 21 -23 - This is true, but not indispensable for the use of DBS for patient monitoring. I suggest a reformulation of the sentence to make it clearer.

We have amended the start of the sentence on line 21 - Ideally, analytical methodologies

  1. Lines 46 – 47 - Although theoretically chromatographic methods are associated with superior specificity, the specificity of FIA-MS/MS is related to the analyte. For some it is specific, for others it isn’t. Please consider reformulating the sentence.

See section below.

 Lines 48 -50 - It is said that specificity in FIA-MS/MS is achieved using SRM, and them it is said that isobaric compounds with a common daughter ion may interfere in quantification. This seams a bit contradictory. On the other hand, SRM is also known for being a more sensitive approach than PS, NL, or other full scan modes. Please consider readjusting the sentence.

We have amended the section starting on line 46 - However, it should be recognised that FIA-MS/MS methods lack specificity. The absence of chromatographic separation means that specificity is achieved solely by the use of selective reaction monitoring (SRM). Consequently any isobaric compound with a common daughter ion has the potential to interfere. A further disadvantage is that the analyte(s) of interest is not separated from the sample matrix, which can result in non-specific interferences from phospholipids and salts.

  1. Line 81 - In the section DBS specimen collection devices, please consider including considerations on how to store the filter paper cards until they are used as well on the expiration date.

We have inserted a sentence at the end of line 90-93 - Filter paper collection devices should be stored as recommended by the manufacturer to ensure that the results of analytical testing are not affected. In addition these collection devices should not be used after the expiry date printed on the filter paper section [4].

  1. Line 163 - It is said that rapid drying improves stability. Please consider adding that it should not be dried on heat sources.

We have amended the sentence on lines 167-168.

  1. Line 223 - It is said that a 3.2 mm punch yields for 3.0μL of sample. This is influenced by many factors, as very well said during the text, playing haematocrit a key role. Please consider indicating a volume interval or referring the value of 3.0μL as an average.

We have amended the sentence on line 228 - A 3.2mm sub-punch is widely considered to yield on average 3.0μL of sample, ….

  1. Line 228 - Please consider including the key importance of the signal/noise ratio.

We have amended the start of the sentence on line 230

 The limit of detection (LOD) of an assay is determined by using the analyte signal to noise ratio (S/N) and is defined as the lowest analyte concentration where its signal can be reliably distinguished from the background instrument noise. Whilst, UPLC-MS/MS instruments are becoming progressively more sensitive, it should be recognised that there is a trade-off between analyte sensitivity and background noise (as the analyte signal increases so does the background, i.e. the S/N ratio decreases). It is therefore important to optimise assays to increase the S/N ratio and therefore improve analytical sensitivity.

The limit of detection (LOD) of an assay is determined by using the analyte signal to noise ratio (S/N) and is defined as the lowest analyte concentration where its signal can be reliably distinguished from the background instrument noise. Whilst, UPLC-MS/MS instruments are becoming progressively more sensitive, it should be recognised that there is a trade-off between analyte sensitivity and background noise (as the analyte signal increases so does the background, i.e. the S/N ratio decreases). It is therefore important to optimise assays to increase the S/N ratio and therefore improve the analytical sensitivity.

  1. Line 311 – 314 - It is said that SUAC requires hydrocloridric acid for elution and ethylacetate for extraction and then that SUAC can be extracted using acetonitrile-water-formic acid containing hydrazine. This seams contradictory. Please consider reformulate the sentence.

 We have amended the start of the sentence on line 321 - Whilst amino acids and acylcarnitines can be extracted from DBS using 80% methanol, SUAC, cannot and requires a different extraction method. One reported method requires the use of hydrochloric acid for elution and ethylacetate for extraction [35]. However, more recently it has been shown that SUAC, along with amino acids and acylcarnitines can be extracted using an acetonitrile-water-formic acid mixture containing hydrazine [36].

  1. Line 343 - There are CRM for some analytes. Please reformulate.

Whilst there are commercially available CRMs for some of the analytes (amino acids) in aqueous liquid form – there are currently no DBS CRMs available commercially. We have amended the sentence (now on line 353) to - A further limitation to the utility of DBS specimens for monitoring IMD patients is the lack of commercially available matrix matched certified reference material (CRM) for the various analytes in DBS specimens on which to standardise laboratory tests.

  1. Line 351 - In this section please consider adding one of the problems when preparing QC samples, that is haemolysis caused by the addition of non-matrix volumes.

 We have amended the sentence now on line 372 - 373 It is recognised that overall the non-matrix spike volume should be <5% otherwise the blood sample matrix is compromised as this can affect the Hct level or even cause haemolysis and therefore sample homogeneity.

  1. Line 404 - It is said that patients should be monitoring using DBS. This is not stated in ref 49. Although with advantages, maybe “should” is a bit too much. Please consider changing it.

We have amended the sentence now on line 423 by deleting the word should

  1. All document - As it is said in the text, one of the main factors influencing the quantification of analytes in DBS is haematocrit. In NBS applications is assumed that it is more or less constant in the first few days of life. But in older infants and adults this is not necessarily true, namely in decompensation/sick periods. So it may be worthy to stress that when using DBS for patient follow-up, the results should be affected by the values of average aged match haematocrit and that DBS are not the best choice for many of the hospitalized patients (under intravenous fluid therapy, decompensation periods, etc). In my experience, the idea that DBS are not the best option in every situation is difficult to assimilate for many heath professionals.

 We have inserted a sentence on line 480 to address this very valid point raised by the reviewer –

Hct varies significantly not only with age and gender, but can also vary significantly during periods of decompensation or as a result of hydration status (i.e. patient may be de-hydrated or be receiving intravenous fluid therapy) and disease states (e.g. liver disease). It is important to recognise that patient results may vary significantly and that analysis of DBS specimens may not be the first choice of specimen in certain situations.